# Spatial Evolution Analysis and Spatial Optimization Strategy of Rural Tourism Based on Spatial Syntax Model—A Case Study of Matao Village in Shandong Province, China

**Xiaonan Qin [1,2,\*], Xueting Du [1], Yue Wang [1] and Lina Liu [1]**

[1]   Business School, Shandong Normal University, Jinan 250399, China
[2]   Geography and Environment School, Shandong Normal University, Jinan 250014, China
[\*]   Correspondence: sdqinxn@126.com

**Abstract:** As the tourism industry is embedded in the countryside, space, which is an important means of production of modern industries, has undergone significant changes in the models and trends of spatial transform. This paper aims to analyze the development pattern of rural economic and social structure driven by the tourism industry. Based on the spatial syntax model, it takes Matao Village of China as the research sample and decomposes the evolutionary process of rural space transform. Firstly (1) in the self-driven development stage, the rural space presented a polar core development trend with high intelligibility and obvious agglomeration. Secondly (2) in the tourism-driven development stage, firstly, the driving force of the tourism industry was relatively weak, whereas the traditional spatial core still occupied an absolute advantage in the whole village space. However, with further development, the spatial scope of high Integration and Choice Values has expanded, resulting in multiple spatial development agglomeration centers. Based on this, it is proposed that a village with a tourism industry should integrate village spatial development planning with landscape style, focus on the regeneration and development of the old space core, and strengthen the integration and correlation of the old and new space clusters.

**Keywords:** space transform; space syntax; driven by tourism industry; space optimization

## 1. Introduction

With the gradual adjustment of global economic and industrial structure, the constant intensification of urbanization and the promotion of informatization, rural spaces are facing a critical period of aggravated change, reconstruction and reorganization [1]. The strategy of "developing multiple forms of appropriate scale management" and "promoting the integrated development of the primary, secondary and tertiary industries in rural areas" promoted the transformation of the rural spatial development mode from a single path and homogeneity to multiple development and heterogeneity. The social capital, state power, foreign tourists and other complex interest subjects brought by rural tourism and the various modernity and mobility forces involved behind them have rapidly and forcefully flooded into rural spaces [2]. Those elements promote the functional change and structural reorganization of rural space and are becoming the core driving force to drive the transformation of rural space rapidly and forcefully [3]. In addition, relative closure of rural space has been broken, and the distribution, organizational form, carried function and transformation pattern of space have all changed significantly [4,5]. Driven by the tourism industry, rural settlement space has focused on the development trend of consumerization and commoditization, exploring differentiated space steering paths and presenting a variety of scenarios, which led to many new industries, new forms and new business models [6,7].

Space as a whole enters into the production model of modern tourism: the land, the underground, the air, and even the light are incorporated into the productivity and

products [8]. Under the impetus of capital injection, industrial adjustment and institutional innovation, spatial production has become the basic driving force and logical development core of the spatial pattern reconstruction of tourism areas [9]. The equilibrium of rights and interests has been oriented in the study of tourism space production, characterizing the justice appeal of tourism industry development. The study of tourism space production is divided into three main themes. (1) Spatial distribution of production materials of the tourism industry. Research on the distribution of the tourism industry [10–12], travel route optimization [13,14] and the layout of transportation facilities [15] to measure the accessibility of the tourism industry, analyze the opportunity equality, process the fairness and result compensation of tourism behavior and recognize the spatial justice of tourism industry distribution. Some scholars have also introduced the theory of spatial transform to analyze the problem of tourism industry distribution. Yi Y. and Zhao J.L. (2017) [16] discussed the transfer and game of land development rights and interests in regulating rural tourism planning from the framework of "space production", and pointed out that the development of rural tourism resources and industrial layout are essentially the formation process of tourism field in the reorganization of land interests. With the appreciation of land and the distribution of such appreciation income, conflicts and contests of power and interests are bound to occur [15]. (2) The spatial distribution of economic production. Relative research explores the pattern evolution of micro landscapes such as urban communities and traditional rural communities, and combed the changes of spatial value driven by the tourism industry [17,18], the replacement of space function [19,20] and the transformation of public space [21,22] in order to interpret the root basis of the distribution of rural tourism rights and interests. It is worth noting that the importance of space power is gradually highlighted [23,24] as an entry point to explore the balance of tourism space production. Sun J.X. (2014) [25] explored the phenomenon and characteristics of spatial reproduction in tourism communities and discussed the spatial representation of community residents against strong power subjects in daily life. Huang T (2016) [26] analyzed the impact of the spatial pattern of tourism on the realization of tourism interests and analyzed the ability of rural residents to obtain industrial benefits.(3) The balance of rights and interests in cultural space. Tourism space needs to become a projective space infused with social emotions and specific forms of social organization and needs to be reconstructed into a meaningful social and cultural entity through subjectivity, thus enhancing the local identity and attachment of local residents [27] (Guo W., 2020). The balance of rights and interests of cultural space lies in the representation of traditional culture and the integration of foreign culture. Researchers identify cultural space production from multiple subjects [28,29], local cultural value changes [30,31], the influence of foreign culture [32] and other aspects as the entry point to analyzing the change of cultural value and the evolution of representation spaces driven by the tourism industry [33,34] to coordinate and integrate multicultural spaces. Huang X.B. and Sun J.X. (2019) [35] pointed out that the theory and perspective of spatial transform can provide a theoretical breakthrough for the myth of traditional village tourism development, and clarifying the spatial rights of villages is a prior issue and a basis for research in this field.

Under the background of tourism development, the livelihood of local residents has undergone great changes, and in some places, they have transformed from traditional "farmers" to modern "citizens" [36], making the livelihood model of farmers stratified and spatially polarized [37–39]. As an external force, rural tourism enters rural communities, inevitably causing strong disturbances to the rural economic structure, social culture, resource allocation, and ecological environment, thus bringing multiple impacts on the livelihoods of farm households [40]. It not only affects the growth and development process of rural space, giving it a distinct core-marginalization character [41,42] but also promoted the change of farmers' traditional livelihood activity space and life activity trajectory [43], thus promoting the reconstruction of farmers' living space [44]. However, rural tourism does not always positively improve the livelihood of farmers but is related to the distance

of farm households from the boundaries of tourist areas, the way the farmers turn to the space and their dependence on geographical space [34,45].

The tourism industry is the driving force of rural space development, and this driving force will also be accompanied by the transformation of rural production space and living space. For this shift in rural space, most of the current studies focus on the conceptual cognitive stage [46,47]—from analyzing relevant theories and concepts and farmers' wishes and behaviors [48,49], to analyzing the spatial shift driving the mechanism of rural tourism development [50–52], spatial circulation mode and spatial pattern [53–55]—so as to further explore the interaction between rural production and living space, rural landscape and social relations. Then, subsequent research focuses on the field of the reconstruction of the rural production and living space [12,56,57]. Most scholars have fully affirmed the positive role of the tourism industry in rural space construction, and put forward the view that spatial transformation can promote the agglomeration effect of tourism industry resources and contribute to the spatial appreciation of the tourism economy [58–60]. In addition, researchers also believe that spatial transformation can improve the spatial value of land and provide a large number of employment opportunities, thereby improving the livelihood of farmers [44,61,62]. In view of the statement that the tourism industry is the driving force of rural space transformation, most researchers hold two basic views. Firstly, rural space should enable rural communities to adapt to the tourism industry "profiting from" rather than "benefiting from" [63]. Those villages propose to encourage residents to participate in the tourism industry [25,64] ensure the residents' right of space construction and cultivate a multi-benefit balance mechanism of rural space resources [39,52,65]. Secondly, the imbalance of spatial development has disturbed the order of rural spatial development. For example, the emergence of issues related to the "tragedy of the commons" in rural communities has aroused extensive concern among researchers [66]. The researchers pointed out that with the increasing popularity of the commercialization of rural space resources; the competition for rural space has become increasingly fierce. However, for this reason, the ownership of the right to use resources has become vague, the property right transfer mechanism has become non-standard, and the supply of public welfare is frequently short [9,67], which has directly led to the abuse of space resources, public order chaos and other situations [26,68,69].

Therefore, researchers have begun to pay more and more attention to the deconstruction analysis of rural space and analyze the evolution law of rural space from the perspective of spatial elements and element relations. Among them, the theory and method of space syntax is to abstract space elements and space structure, separate space from the economic system and the social system, return to the quantification and analysis of space itself, and then further interpret the evolution of the economic and social system from the analysis of changes in space structure [70]. Then, researchers use the space syntax model to analyze the architectural layout, residential space, street form and other aspects, as well as the internal laws and expansion rhythm of the development of the space system [71,72]. Moreover, in the study of rural space, Chinese researchers also use a space syntax model to describe the spatial morphological characteristics of rural tourism, further analyze the mechanism and development potential of village space and accurately interpret what the tourism industry injects into rural space and the optimization proposal of industrial space construction [43,73]. To sum up, the tourism industry is driving the structural change of rural space elements, driving the commercialization of rural space. The diversification of elements, the compounding of functions and the differentiation of structures have been driving the transformation of rural space to a new development mode and direction. The research field has only recently started. Researchers pay more attention to the definition of the concept of rural space and the recognition and attention to the spatial rights and interests of the multi stakeholders who inspire rural tourism. However, research on the deconstruction and spatial transformation of rural space is still in the stage of enrichment. The rural spatial syntax research becomes a more effective way to deconstruct spatial elements and understand and analyze spatial steering drivers.

To address the research gap in existing studies, this study focuses on three research questions. Firstly with the embedding of the tourism industry, how does the rural livelihood space evolve and develop, and how does the structure of original living space and labor space transform into the tourism industry space? Secondly, what is the best method to realize the analysis and cognition of spatial elements and to analyze the distribution logic and law of rural spatial elements? Thirdly, what is the best method to combine the abstract spatial elements with the process of economic and social development, so as to recognize the influence of the construction of spatial elements on the rural economy and society? On account of the evolution of spatial morphology, this paper analyzes the development pattern of rural economic and social structure driven by the tourism industry based on the evolution of spatial morphology. The research objectives are as follows: (1) deconstructing the spatial elements and structure and describing the historical process of rural spatial and temporal evolution in detail quantitatively; (2) combing the characteristics cognition, model induction and evolution history of spatial transformation driven by the tourism industry in detail; and (3) analyzing the development law of rural economic and social structure driven by the tourism industry and exploring the optimization path of rural spatial transformation by the evolution of spatial form.

## 2. Research Area and Research Methods

### 2.1. Research Area and Data Preprocessing

Matao Village in Shandong Province, China, is located in the southern end of Wande Street, Changqing District, Jinan City, Shandong Province. Located in the northeast of China, Matao Village belongs to the Taishan Mountains and has mountain landscape, tea garden landscape and folk culture as its main tourism resources (as shown in Figure 1). In 2019, Matao Village was awarded the title of "the China national key rural tourism village", becoming the benchmark and leader of the tourism industry in Shandong Province. There are two important time nodes in the development of Matao Village (as shown in Figure 1).

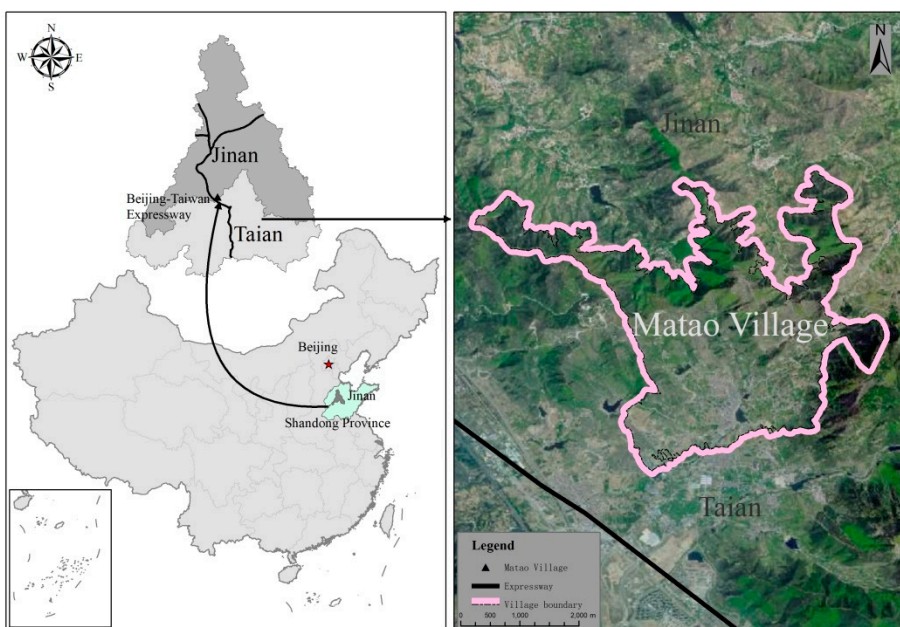

**Figure 1.** Geographical location of Matao Village.

Firstly, in 2008, the tea industry park was introduced, and the development of the tourism industry began. During this development process, Matao village did not yet possess a unified tourism management department. The development of its tourism industry depended on the independent participation of its local residents. A group of elite personnel in the tourism industry gradually emerged. Small-scale tourism industries and

space construction were conducted. Some small rural hotels and a tea entertainment park were built. The space construction of its industry came mostly from the residents' own investment, the industrialization of all their houses and an increase in the tourism service functions. Their industrial participation behavior played a beneficial role in inspiring the surrounding residents.

Secondly, in 2015, it was awarded the title of "Beautiful Leisure Village in China". The tourism industry developed in an all-round way, and tourism facilities such as water entertainment areas, mountain homestay areas and tourist distribution centers were built. Nowadays, the tourism industry has become the pillar industry of Matao Village. In 2019, the number of tourists received was 650,000, and the tourism income was CNY 19.8 million, which realized the transformation from rural tourism poverty alleviation to tourism enrichment. In this process, Matao Village established a unified tourism industry management department—"rural tourism cooperatives", which adopted the management mode of residents' participation and collective decision-making. The construction of the tourism industry space had gradually changed from individual investment to collective fund-raising and unified planning and construction. At the same time, Matao Village also began to gradually introduce foreign investment. The village used space transfer leasing and space cooperation and development to enhance its rural space construction capacity. The participants of rural space production also changed from local residents to tourism industry cooperatives and foreign investors.

Due to the impact of COVID-19, the development of the tourism industry in Matao Village was at a standstill from 2020 to 2022. Restricted by the epidemic prevention policy, the number of tourists dropped significantly, and the space development of the tourism industry was also at a standstill. With the optimization and adjustment of the Chinese epidemic prevention policy, this situation has been gradually alleviated, and the rural tourism industry has been in a state of recovery since 2023.

According to the availability of research data and the characteristics of development stages, this study considers 2003 to be the starting point of research and selects four years of satellite remote sensing images in 2003, 2008, 2015 and 2020 as the research data map (Data from: https://www.resdc.cn/, accessed on 15 May 2003; 2 June 2008; 27 May 2015 and 24 May 2020). The axis model of space syntax is drawn by AutoCAD software. In the process of data vectorization, this paper takes the rural village as a whole continuous space, which means that local residents and tourists can follow the natural action law to freely travel within the space. Based on this principle, we remove the closed and confidential area and the internal road of the scenic spots and residents' courtyards, so that all the lines on the axis model could be set to the same weight to ensure the homogeneity and continuity of the sample space.

*2.2. Basic Principles and Morphological Variables of Space Syntax*

Oriented to the development process of Matao Village, we find that the development of the rural tourism industry basically focuses on space construction and space expansion. The breadth and ability of space construction determine the level of rural tourism industry development. According to the stage characteristics of space production, we can more clearly sort out the stage characteristics of the development of the rural tourism industry. Based on the research and analysis of the characteristics in different stages of development, the recent studies can also explore the problems of space production of the tourism industry and put forward strategic suggestions for space optimization. Therefore, the research herein takes rural space as an important factor of tourism industry production and introduces the spatial syntax model to analyze the current situation of spatial production in different development stages.

Space Syntax is a theory and method based on abstract cognition of space, quantifying spatial components and analyzing the relationship between spatial comprehensive systems and local construction. Based on the model, relative research can discriminate the logic and rhythm of spatial distribution and combine spatial construction with the economic

and social system, aiming to reveal the causes, mechanisms and development trends of the distribution of economic and social elements [70]. The core of space syntax is the abstract induction and quantitative analysis of the relationship, and the presentation and expression of complex and abstract spatial components in a clear way. Through the establishment of axis analysis model, line segment analysis model or horizon analysis model, the spatial form is analyzed [73]. Space Syntax separates the spatial elements from the economic and social system for independent analysis and judgment, and then in turn analyzes the driving effect of spatial structure on the background causes and development trends of the evolution of the economic and social system. The specific spatial form evaluation indexes include Integration Value (Global Integration and Local Integration Value), Choice Value, Intelligibility Value and Synergy Value. Based on the establishment of the axis model and line segment model, Depthmap 1.0 software is implied to calculate its spatial form evaluation index. This study constructs a spatial syntactic model that takes spatial morphological structure as the main body of research and analyzes the development pattern of rural economic and social structure by the evolution of spatial morphology. The specific research ideas and research models are as follows (as shown in Figure 2).

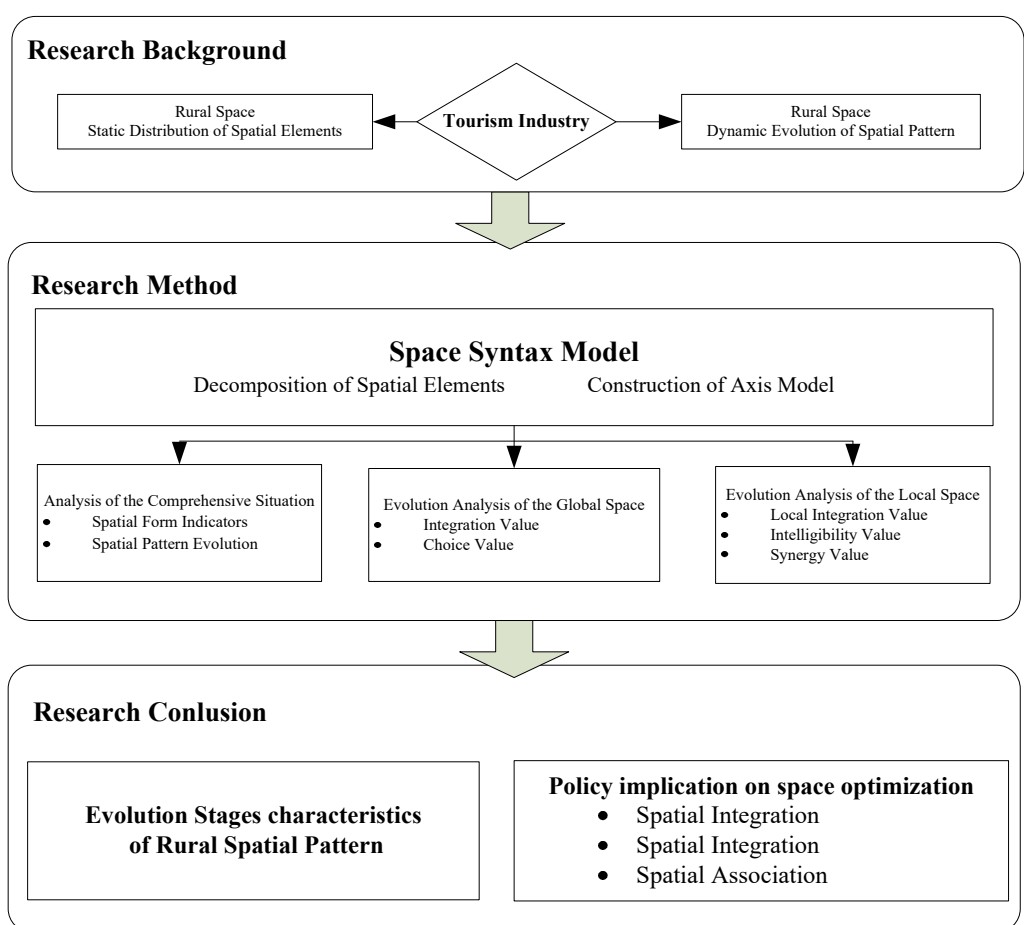

**Figure 2.** Research route.

(1)  Integration Value

Integration Value describes the degree of spatial agglomeration, measures the degree of association between individual space and other spaces and reflects the texture structure and functional distribution of the overall space. The highly integrated space is a block with concentrated distribution of functions, which indicates that the individual space has high agglomeration centrality and spatial accessibility. Spatial integration is divided into two types: Global Integration Value and Local Integration Value. Firstly, Global Integration

Value refers to the close degree of connection between individual space and all unit spaces in the whole space. Secondly, Local Integration Value refers to the degree of closeness of spatial interaction between individual space and its nearby measurement radius. The measurement radius of local integration can be measured by topological units ($R$ = 1, 3, 5, 7, etc.) or metric units ($R = n$ meters).

(2)    Choice Value

Choice Value refers to the probability that a road becomes the shortest path from all spaces to all other spaces. Choice Value is used to describe the extent to which this street is part of the shortest path, reflecting the individual's spatial efficiency and development vitality. The higher the Choice Value, the higher the probability of the path as the core through the road, the more able it is to gather more people, and the more likely it is to become an effective development space.

(3)    Intelligibility Value

Intelligibility Value measures the similarity between the individual spatial texture and the overall spatial structure, using the linear correlation coefficient ($R^2$ between the connectivity of the individual space and the degree of integration under the radius $n$, reflecting the identifiability, coordination and rationality of the overall spatial form. The higher the Intelligibility Value is, the stronger the correlation between the individual space and the global space is, and the higher the consistency is between the perceptible space and the overall unknown space.

(4)    Synergy Value

Synergy Value refers to the correlation coefficient ($R^2$ between integration in individual space and global integration. This index is similar to the definition of intelligibility, which represents the similarity between the single space and the whole space and reflects the characteristic consistency and equilibrium of the space system.

## 3. Evolution Analysis of Spatial Transform in Matao Village Based on Space Syntax Model

*3.1. The Comprehensive Situation of Spatial Transform in Matao Village*

After 18 years of development, the evolution of the spatial pattern of Matao Village has shown significant stages. In different stages of development, the development trend of rural space shows various characteristics (as shown in Figure 3). Firstly, self-driven development stage within the village from 2003 to 2008: the building area increased by 15%. Its space development was mainly based on the needs of villagers' production and lives, and the path of space commercialization was relatively single. The space construction was mostly around the residential agglomeration area in the middle of the village, the flat terrain and the surrounding area of the core road of the village. The types of space utilization were mainly based on the expansion of homestead and cultivated land area, showing fragmentation and randomness. Secondly, the 2008–2020 tourism industry-driven development stage: the construction area increased to 24.5%, and the rural space from the center of agglomeration became a multi-center, multi-theme situation. Road network density increased significantly, forming a comprehensive coverage of the village space. The spatial turn no longer revolved around the flat terrain and convenient traffic space area but tended to the mountain area and water coast with superior tourism value. The spatial transform was no longer limited to the self-demand of villagers but was divorced from the control and dominance of villagers. It deviated from the residential agglomeration area and tended to be more rural space with tourism industry value and single spatial rights and interests.

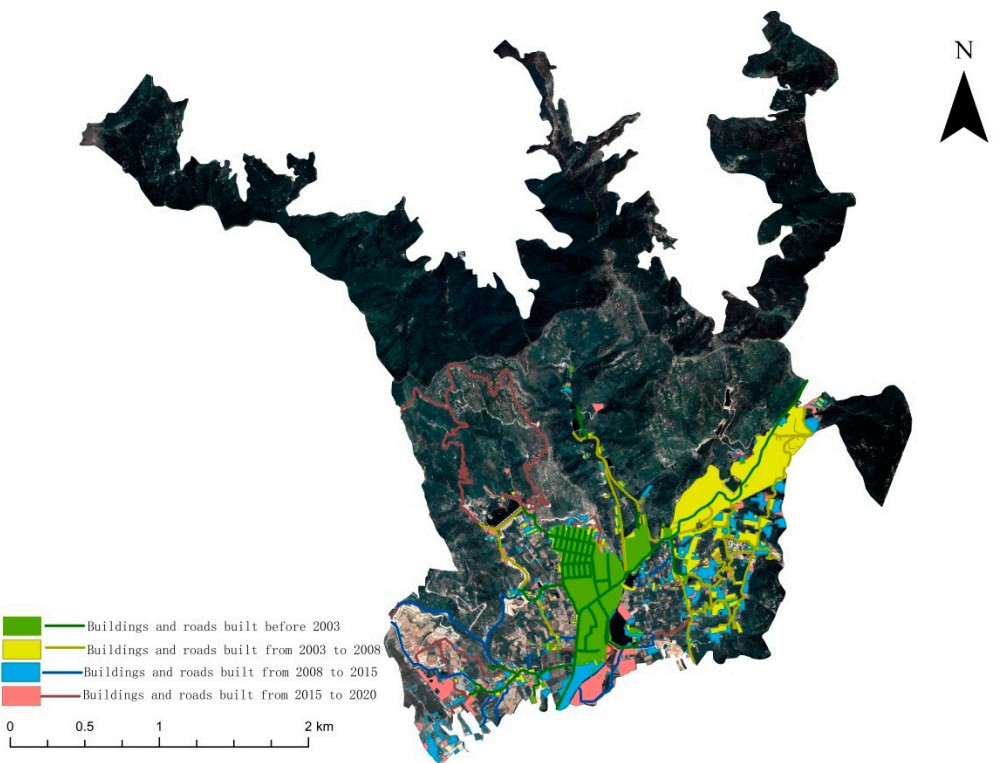

**Figure 3.** The spatial evolution process of Matao Village.

Based on the analysis of the spatial form indicators, the spatial form of Matao Village presents the following characteristics (Table 1). Firstly, the mean of the Integration Values and its coefficients of variation show a steady downward trend. The Integration Value is the centrality of spatial units and measures the potential of spatial development, which reflects that the spatial polarization effect is differentiating, forming a diversified central agglomeration area, and the spatial scope of the effective activities is gradually dispersed and expanded. Secondly, the mean of Choice Value increases significantly, and its coefficients of variation also increase significantly, indicating that the spatial difference has increased gradually. The increase of Choice Value reflects the effectiveness of village spatial planning, which makes all kinds of resource elements in the village effectively connected. The coefficients of variation are significantly improved, reflecting the imbalance of village space development, and the spatial turn is more inclined to the space construction with more industrial value. Thirdly, street network density shows a downward trend. Especially between 2003 and 2008, its value falls by 21%, and then the downward trend slowed down. China's rural space construction is mostly led by rural road construction to build the basic framework of space transformation. Matao Village has been actively involved in the construction of rural roads since the beginning of space construction, covering the entire village as much as possible and building a basic framework for subsequent spatial development. Nowadays, the road network architecture of Matao Village is basically completed, and the decline in street network density has gradually slowed down.

**Table 1.** Matao Village spatial form indicator.

| Year | | Integration Value | Choice Value | Street Network Density |
|---|---|---|---|---|
| 2003 | Mean | 0.33 | 7730.22 | |
| | Standard deviation | 0.1 | 11489.42 | 63.32 |
| | Coefficient of variation | 0.29 | 1.49 | |

**Table 1.** *Cont.*

| Year | | Integration Value | Choice Value | Street Network Density |
|---|---|---|---|---|
| 2008 | Mean | 0.28 | 21133.46 | |
| | Standard deviation | 0.07 | 35158.64 | 49.98 |
| | Coefficient of variation | 0.27 | 1.66 | |
| 2015 | Mean | 0.22 | 49342.12 | |
| | Standard deviation | 0.06 | 88711.89 | 45.66 |
| | Coefficient of variation | 0.23 | 1.8 | |
| 2020 | Mean | 0.23 | 57521.15 | |
| | Standard deviation | 0.04 | 107133.4 | 43.94 |
| | Coefficient of variation | 0.18 | 1.86 | |

*3.2. Evolution Analysis of the Global Space of Matao Village*

3.2.1. Evolutionary Analysis Based on Spatial Global Integration Value

Based on the analysis of the evolutionary map of the Integration Value of Matao Village, the change of Integration Value shows obvious stages, and the specific spatial structure changes show the following characteristics (as shown in Figure 4). Firstly, the axis with high integration of Matao Village is mostly centered on the traffic arteries connecting the central and western parts of the village and expands as the center. The axis with a relatively low degree of integration is distributed in the rural marginal areas, especially in the western region centered on farming and the northeastern region dominated by high mountains. Secondly, the area with high integration is always concentrated near the residential area. Although in the rapid development stage of the tourism industry from 2015 to 2020, construction of the roads and buildings is no longer carried out around the residential agglomeration area, the Integration Value of the area is still improving. This is because Matao Village has built infrastructure such as tourist reception centers, hotels and parking lots around the residential area, and the village's main roads are all around the residential area. Whether tourists choose to go to the water entertainment area or mountain homestay, they must go through the core road, which extended from the residential agglomeration area. At the same time, Matao Village focuses on the development of rural folk culture and the red culture, so that the residents' gathering area also has the corresponding tourist value and becomes the key block of tourism industry space. Thirdly, the integration growth area shows a trend of east–west expansion, and the integration degree of the homestay agglomeration area in the high mountains is the most significant, reflecting that this area will be the core area of the village space construction in the next few years.

3.2.2. Evolutionary Analysis Based on Spatial Choice Value

Based on the analysis of the evolution map of the Choice Value of Matao Village, the evolution of the Choice Value has the following characteristics (as shown in Figure 5): in the stage of self-driven development within the village, the change of the Choice Value is relatively small, and the main changes were reflected in the extension of the core road and the increase of the density of the eastern road network. In the stage of tourism industry-driven development, the change of Choice Value begins to intensify and shows a multi-center common development trend: on the one hand, the core road continued to extend to the northeast, becoming the longest village road, which played an important role in connecting the overall spatial skeleton. On the one hand, the eastern residential agglomeration area, the western water entertainment area and the tea garden cultural area all show a high Choice Value in turn, reflecting the transformation of tourism to the overall rural space and promoting the diversified development of rural space.

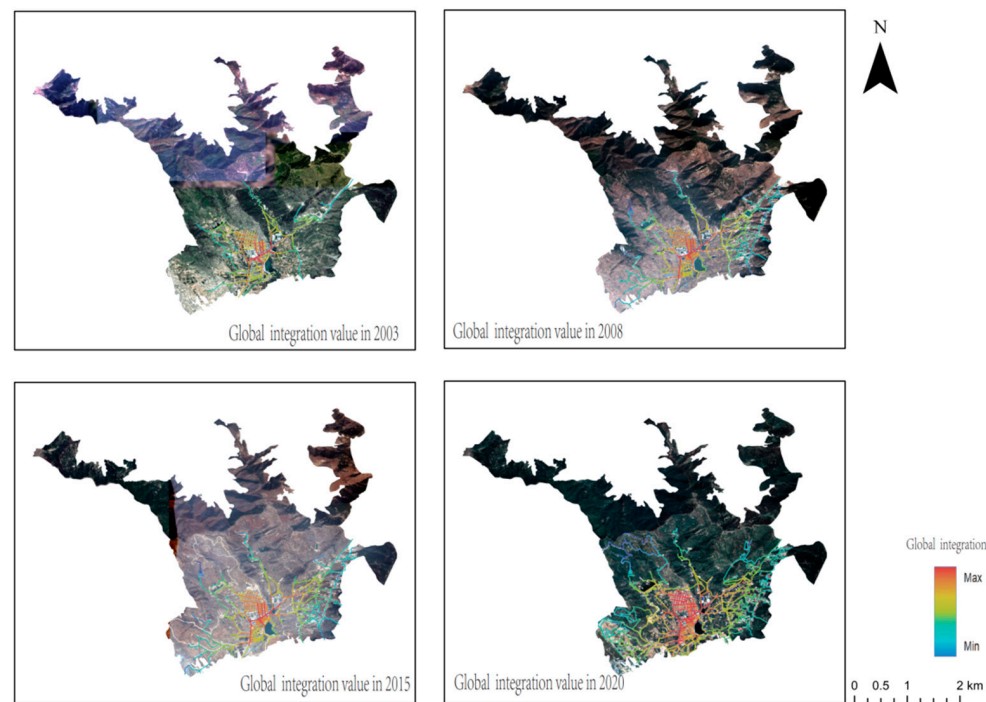

**Figure 4.** Evolution process of spatial global Integration Value in Matao Village.

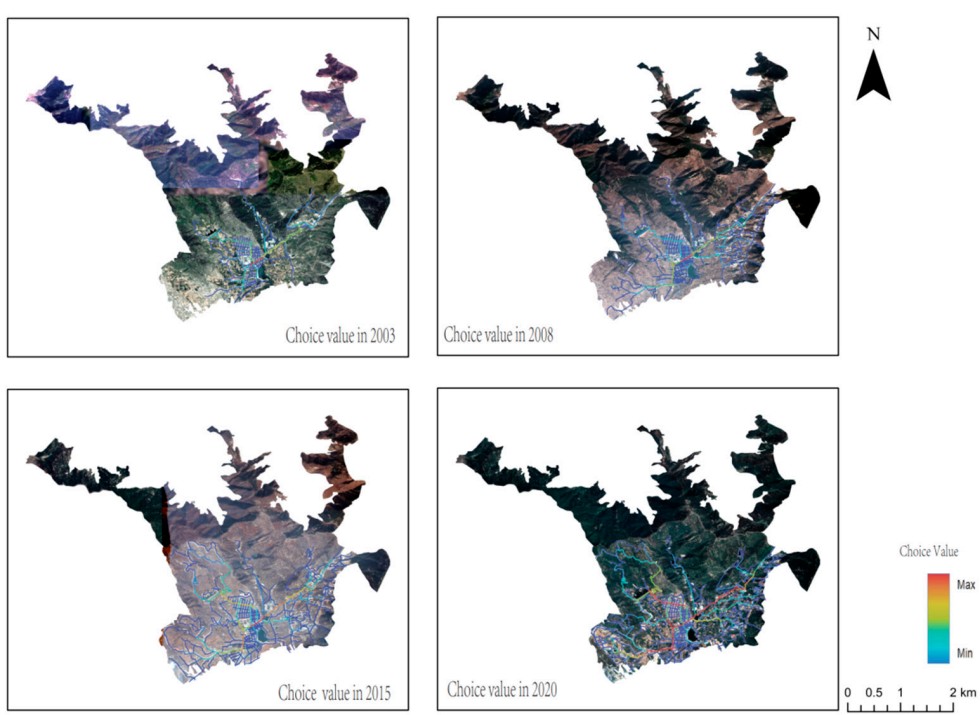

**Figure 5.** Evolution Process of spatial Choice Value in Matao Village.

### 3.3. Evolution Analysis of the Local Space in Matao Village

To study and build the spatial line segment model of Matao Village, first, we take the space measuring radius of 2000 m as the upper threshold, and take 500 m as the span for experiment. After several operations on the model, the research results show that when the measurement radius is 500 m, the measurement radius is too low, and the value of the long line segment is significantly improved, thus reducing the importance of the branch road. When the measurement radius is 1500 m and 2000 m, the change of local integration is not very obvious and is closer to the map of global integration. Therefore, the research

selects a measuring radius of 1000 m to analyze the local space components and uses the local Integration Value, the Intelligibility Value and Synergy Value to analyze the relevance and integration of local space and the overall space.

### 3.3.1. Evolution Analysis Based on Local Integration Value

According to the map of local integration with a measuring radius of 1000 m (as shown in Figure 6), the local spatial pattern of Matao Village mainly presents the following characteristics. Firstly, in the stage of driving development within the rural area, the spatial distribution of local integrity shows a relative state of spatial agglomeration and path dependence, which is mainly concentrated on the core roads extending from the residential area. At this stage, the coverage area of rural roads is on the rise, but the density with the road network is gradually decreasing, and the distribution of the overall local integration of the village has not changed significantly. These situations clearly reflect that the actual activity space of people is relatively concentrated and does not spread to the surrounding areas with the space development. Secondly, when the tourism industry is in the development stage, the local Integration Value shows a trend of multi-point growth, which reflects that rural space cultivates multiple local space cores. Therefore, the core roads around the tourist distribution center and the home-stay cluster area become the new space development pole.

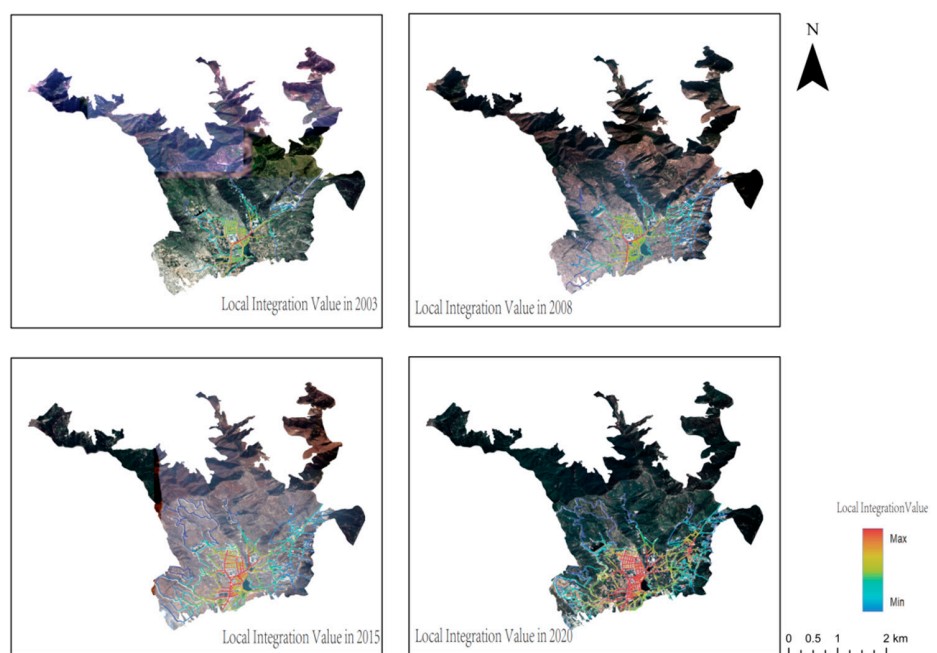

**Figure 6.** Evolution process of local integration in Matao Village.

It should be noted that there is no good correlation between the spatial polar core of the home-stay cluster area and the traditional spatial cluster area, and the local Integration Value of the space between them is relatively low. In the field survey of the research team, we found that Matao Village dose not strictly prohibit non-home-stay tourists from entering the gathering space, but because the home-stay gathering area is mainly concentrated in the high altitude mountains, the external traffic roads are single and the accessibility is poor. However, an independent and convenient road network has been formed in the home-stay cluster area, and a landscape system different from the traditional village style of Matao Village has been cultivated. In this way, home-stay tourists can obtain convenient accommodation, catering, recreation, entertainment and other tourism products in the cluster area. However, the home-stay cluster space is not closely integrated with the rural tourism industry of Matao Village but is independent of the rural space system in the form of "industrial enclave".

### 3.3.2. Evolution Analysis Based on the Intelligibility and Synergy Value

The spatial Intelligibility Value of Matao Village is relatively low ($Ri^2 < 0.5$) and shows a decreasing trend. In 2003, the Intelligibility Value was 0.325, but in 2008–2020, the intelligibility gradually decreased to 0.309, 0.299 and 0.287. The gradual decline of the Intelligibility Value reflects that the intelligibility of the space of Matao Village is not ideal, and there is a lack of unified space development concept and planning. In addition, when foreign tourists enter the village space, they need a long time to adapt and explore, and it is difficult to perceive the spatial form of the whole village through the local space. The rural space, which further deviates from the morphological norms and landscape styles of traditional rural space, is more inclined to serve the development needs of the tourism industry and then cater to the cognition and habits of foreign tourists.

According to the analysis based on the Synergy Value map with a measuring radius of 1000 m (as shown in Figure 7), Matao Village has a relatively high value of synergy ($Rs^2 > 0.5$), which reflects that the rural space has a centralized core axis, people's travel and activity spaces are relatively concentrated, and its spatial steering also shows a significant path dependence. Furthermore, the degree of synergy is also decreasing year by year, with the Synergy Value being 0.708 in 2003 and gradually decreasing to 0.695, 0.603 and 0.506 from 2008 to 2020. With the spatial production process of the tourism industry, the spatial development of the tourism industry has been strengthened, and a large number of tourists have poured in, resulting in a diversified and large number of social economic spatial activities. The interaction and correlation between different rural spaces is also enhanced, forming a new core axis within each spatial unit and building a communication line between spatial units. At present, the "concentric circle" spatial expansion mode of Matao Village, with the residential cluster as the core, has gradually been weakened. On the contrary, the diversified spatial polar core around tourism elements and industrial demand has gradually formed and begun to transform from a single spatial polar core to a diversified spatial cluster. The development of each local space is relatively independent, and a unified spatial development plan has not been formed yet.

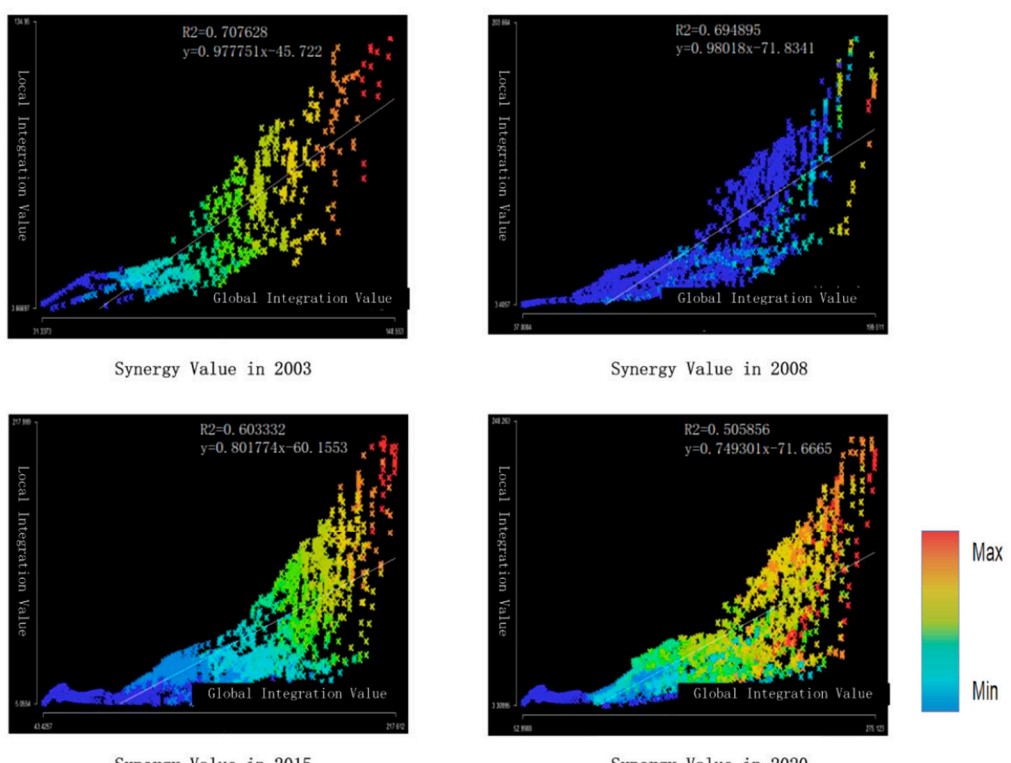

**Figure 7.** Evolution process of spatial Synergy Value in Matao Village.

## 4. Evolution Stages Characteristics of Rural Spatial Pattern in Matao Village

Based on the analysis of the evaluation indicators of the above spatial syntax model, the research summarized the process of spatial transformation of Matao Village. The specific spatial pattern research is shown in Table 2:

**Table 2.** Evolution phase of spatial transform in Matao Village.

| Space Transform Stage | | | Time | Evolutionary Characteristics |
|---|---|---|---|---|
| Rural self-driven development stage | | | Before 2003 to 2008 | 1. Space presents the development trend of polar nucleus 2. The effective active space shows a "linear" gathering trend, with a large range of decline towards the periphery 3. The Synergy Value is relatively high and the concentration is obvious. |
| Tourism industry driven development stage | Sub stage 1 | Point embedding and Line embedding | From 2008 to 2015 | 1. The spatial driving capacity of the tourism industry is relatively weak. 2. The old core still occupies an absolute advantage in the space of the whole village and fails to realize the change of village centrality 3. With the extension of main roads as the main form, the basic framework has completed the village road network system. |
| | Sub stage 2 | surface embedding | From 2015 to 2020 | 1. The spatial driving capacity of the tourism industry has been comprehensively improved. 2. The spatial scope with high degree of Integration and Choice Value has been expanded, resulting in multiple spatial development cluster centers. 3. The focus of space development has also changed from the southwest to the northeast, gradually breaking the pattern of single core cluster in the village. |

Phase 1: Development stage of rural self-driven spatial transformation (from before 2003 to 2008).

Matao Village followed the traditional Chinese rural development model, and the spatial expansion was more manifested in the expansion of agricultural land and residential land. The rural space presented a significant polar core development model, spreading to the surrounding areas centered on residential agglomeration areas. The distribution space of Integration Value, Choice Value and local Integration Value tended to be consistent, and the coverage was small, mainly concentrated around the core roads extended by residential agglomeration areas. However, Matao Village had also actively carried out the construction of the road network to expand effective production and living space. In spite of this fact, the development trend of space polar core had not been reversed, and the effective active space shows a "linear" agglomeration trend, and the decreasing pole difference to the surrounding area was large. The spatial comprehensibility was relatively high, and the agglomeration was obvious, reflecting that the spatial transformation path of Matao Village at this stage was relatively single, and the spatial production followed the needs and laws of villagers' production and life, forming a significant constraint and standardization.

Phase 2: Development stage of tourism industry driven spatial transformation (including 2008–2015; 2015–2020).

From 2008 to 2015, "point" embedding and "line" extension were the main stage, which was the weak driving stage of the tourism industry. In 2008, Matao Village built the

Northern Tea Garden Plantation Project, officially introducing the tourism industry into the rural space. At this stage, the tourism industry was limited to optimizing traditional farming and production methods, embedding "point" tourism elements into the original space of the countryside and optimizing the functions of the original production space and living space. In general, the role space of the tourism industry was relatively limited, and the core of the old space still occupies an absolute advantage, maintaining strong spatial agglomeration and path dependence, and there was no central change in rural space. At the same time, the spatial development of Matao Village took the extension of the main road as the main form, and the improvement of spatial integration and selection showed a trend of southeast extension, reflecting that the tourism industry also relied on the core trunk road of Matao Village, supported the basic services of the tourism industry, built the basic structure of the industrial space and laid the foundation for the subsequent infill development of industrial elements.

From 2015 to 2020, the development of "surface" extension was the main stage, which would be the comprehensive driving stage of the tourism industry. In 2015, Matao Village established a tourism professional cooperative to promote the transformation of villagers into "housing shareholders", used idle or newly built houses to build home-stay clusters and built tourist distribution centers, water entertainment areas and other faceted space construction projects. At this stage, the tourism industry promoted the comprehensive transformation of rural space. The spatial scope with high degree of integration and selection was increasing, resulting in multiple spatial development agglomeration centers. The focus of village space development had also been transformed from the southwest tea garden cultural area to the northeast homestay cluster area, breaking the form of single-core agglomeration in the village.

According to the above research results, the spatial transformation of Matao Village is in a rapid change stage which mainly focuses on the layout and link of tourism industry elements and has not yet considered the coordination and integration between industrial space and traditional rural space. With the gradual decline of spatial intelligibility and synergy of Matao Village, a diversified spatial polar core and landscape style have emerged, and the "concentric circle" spatial expansion mode has gradually changed into a parallel and independent development trend of multiple spaces. In addition, the tourism industry space tends to be "isolated from neighbors" [74], that is, isolated from the residential area, and its spatial form and landscape style are in sharp contrast with the traditional village space.

## 5. Discussion and Conclusion

### 5.1. Discussion

Compared with other industries, the tourism industry presents a stronger spatial reliance, and the organizational planning of tourism products is actually the reconstruction and reorganization of the tourism production space [2]. When the tourism industry is embedded in rural space, space as an important production material has entered the production mode of modern tourism [44,45], not only accommodating the construction and use of tourism assets that carry them but also as an important production material to realize tourism production and tourist experience behaviors. The rural spatial transform has also detached from the demand-led and villagers' spontaneous mode and turned towards the industrialized and commercialized transform mode.

In the process of tourism-driven rural spatial transform, space, as a means of production, presents strong concealment and deception [12]. It is difficult to achieve ownership at the legal and policy levels, similar to contracted land, homestead and other means of production [24]. It is precisely because of the ambiguity of space ownership that rural inhabitants realize that space begins to be a kind of value capital, and its exchange value is gradually greater than the use value. "Space transform" can bring livelihood support and improvement to themselves, but they still cannot establish their ownership and decision-making power over livelihood space and cannot rely on "space power" to

obtain corresponding "space interests" [55]. In the face of the imbalance and polarization in the rural tourism space, we can no longer rely solely on the understanding game between multiple stakeholders and the normative guidance of rural tourism policies, but it is necessary to lay the theoretical tone of "space as the core means of production" fundamentally. Based on the theoretical perspective of "space is power", we can support rural inhabitants' ownership and profitability of rural space that depends on generations, so as to build a new win–win tourism production space order of "rural tourism industry-social sustainable development".

In the relative research on rural space, researchers have focused on the layout of tourism industry space and have discussed the participation ability of rural residents in the construction and use of the tourism industry space [50–53]. The research of this paper is to sort out the process of the tourism industry driving the transformation of rural space and to dismantle the rural elements by the space syntax model. Based on this, this paper studies and analyzes the law of the embedding and occupation of rural space by the tourism industry, and analyzes the process and mode of all the space loss and transformation of rural residents in the process of space transformation, so as to recognize how the residents' space rights and interests are transferred and marginalized in the process of rural space reconstruction.

Through the case analysis of this paper, we found that the tourism industry's encroachment on the rights and interests of rural space is not by the direct purchase of space use rights or ownership, nor by physical isolation of space, but it is through more invisible space deprivation. Through the establishment of "enclave" economic space, the tourism industry intensifies the differential rent of village space, making the residents' power gradually marginalized and weakened in the process of spatial capital transformation [58,59]. At the same time, foreign tourism investors establish spatial features and spatial patterns different from traditional rural landscapes, guide tourists' consumption tendencies and reduce the tourism market's recognition of traditional rural spatial features and spatial patterns. These methods are also an important way to weaken the ability of residents to capitalize space [55,70,71].

Therefore, in the construction of rural tourism space, residents should be the real owners of rural space to awaken awareness of space rights. The process of spatial transformation driven by the tourism industry should properly follow the sequence and style of the traditional core space in the village; introduce the usual mode and spatial style of residents' production and life into the emerging development industry space; strengthen the connection between the old and new space; maintain the efficient operation of the old and new space; transform the independent industrial space into the usual living space, which is convenient to achieve and has multiple functions; and truly integrate the value of tourists and the tourism industry into the rural spatial pattern.

*5.2. Conclusion*

Based on the theory and research method of spatial syntax, this paper selects Matao Village, a "beautiful leisure village in China" in Shandong Province, as a research sample and constructs the axis and line segment model of spatial syntax to identify the influence of the tourism industry on the spatial transformation of the village.

(1)    From the analysis of the overall spatial pattern, Matao Village's space shows significant stages. In the stage of intra-rural driven development, the rural space shows a strong central concentration and path reliance, with the rural settlement as the core of spatial construction. The spatial integration and selection are spread to the periphery with this space as the center. During the tourism industry-driven development stage, the rural space gradually breaks the trend of single-core development and extends to the tea garden culture area in the southwest and the B&B gathering area in the northeast, presenting a multi-centered co-development trend.

(2)    From the analysis of the correlation between the local formation of space and the overall space, in the stage of driving development within the countryside, the areas

with a high degree of integration are concentrated on the rural settlements and core roads, and the spatial development and construction patterns are obvious and systematic. However, the central space is more different from the surrounding space, forming a significant spatial disconnection. At the stage of tourism industry-driven development, the degree of spatial integration appears in multiple clustering polar cores, which present their own trend of development, and the overall spatial cognition gradually decreases. Additionally, the correlation between multiple spatial polar cores is weak and has not yet been able to effectively drive the synergistic development of the entire rural space.

To sum up, the spatial transformation of Matao Village has a strong directional character, and the spatial form is transformed from a circle-type expansion with central concentration to an extension along the core road growth axis to the east and west. The spatial evolution of Matao Village is from a single pole core to a multi-center and from a continuous central concentration to a discrete stage pattern of multiple differentiation.

### 5.3. Policy Implication

(1) The renewal and development of the traditional core of the village space. The rural spatial transformation often follows the "central concentration" and "path dependence". In addition, the resident agglomeration area in the center of the village has always been the core of spatial production, carrying the functions of the daily life of residents and the gathering of tourists [51,52]. The construction of additional tourist infrastructures, such as tourist distribution centers, catering facilities and parking lots in the area, has made the space even busier and more congested, making it difficult to improve its spatial pattern and architectural form. Moreover, the old and congested residential agglomeration space forms a sharp contrast with the new tourism industry space. Therefore, the village should explore a development model to break the traditional "center-axis" development model, optimize the existing road system, divert villagers and tourists and build a new road dedicated to production and life. The style of old space should be optimized and improved, weakening the differences between emerging tourism space and traditional area. Additionally, it should build the interaction path between tourism space and traditional space and enhance the identity of villagers and tourists to rural space.

(2) The integration and organization of new spatial agglomeration cores. The emerging spatial agglomeration cores and the traditional old cores often develop separately. Moreover, there is a spatial break, reflecting the isolation of the tourism industry development and rural space and depriving the right of rural residents to produce and share the value of the tourism industry together [54,55]. Thus, the village should explore an introduction of emerging tourism into the core of the old traditional space, optimize the value-added industrial space and integrate it with the traditional space and available tourism products, such as tourist stores, residential lodges and B&B museums in the residents' gathering area. In the new industrial space, we should continue the spatial characteristics of the traditional village, break through the spatial expansion mode of "axial growth", and plan new central roads to enhance the correlation of multiple spaces and improve the accessibility of space.

Further research is required. Firstly, the role of spatial form on rural social and economic space will be analyzed by projecting economic and social elements on top of the rural morphological structure. Secondly, the study will try to invert the subject of the study; explore the driving role of the rural spatial transformation by the demands of rural residents, tourism practitioners and tourists and other spatial users of the countryside; and provide relevant strategic suggestions for the optimization of the rural spatial structure. Thirdly, we will try to further mine and use the spatial form evaluation index data measured by the spatial syntax model. These index data would be combined with other relevant economic and social indicators and subjected to comprehensive geospatial analysis methods

to analyze the impact of the spatial construction status of the tourism industry on the rural economic and social development.

**Author Contributions:** Data curation, funding acquisition, writing—original draft: X.Q. and Y.W.; conceptualization, methodology: L.L. and X.D.; methodology, investigation: Y.W. and X.D. All authors have read and agreed to the published version of the manuscript.

**Funding:** Natural Science Foundation of China: 41901169; Shandong provincial Natural Science Foundation: ZR2019BG001; Humanities and Social Sciences Fund of the Ministry of Education: 20YJC790061.

**Conflicts of Interest:** The authors declare no conflict of interest.

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
