# Peer review of "Spatial Evolution Analysis and Spatial Optimization Strategy of Rural Tourism Based on Spatial Syntax Model—A Case Study of Matao Village in Shandong Province, China"

_land, doi:10.3390/land12020317_

Round 1
Reviewer 1 Report
Manuscript title:
Spatial Evolution Analysis and Spatial Optimization Strategy of Rural Tourism Based on Spatial Syntax Model-- a case study of Matao village in Shandong Province, China
Review:
The name of case study, is it Matao or Ma Tao as written in conclusion. Why is there different spelling?
(line 12) “constructs the axis and line segment model of spatial syntax, analyzes its spatial form and evolution process, and puts forward the strategy of spatial optimization”. A figure drawing these axis, spatial form and optimization in figure(s) overlay with satellite image of Matao village would be helpful to make better understanding the materials in this manuscript.
The audience of this journal may not know where is the location and geographical scope of Matao village in Shandong Province. More descriptions and pictures of this region are indispensable: its location in China nation, as well as its tourism development condition (number of tourists, stakeholders involved in spatial and temporal evolution, etc). Argumentative writing is needed on why this geographical scope must be analyzed using the space syntax concept. The writing in (line 161) “2.1. Research area and data pre-processing” is not adequate to enable the reader to understand this region.
Refinements of figures space syntax methods and mapping levels are needed, to include more recent and detailed layers. This could also involve modeling spatial evolution analysis of the phenomena under study, such as tourism development diagrammatic picture(s).
Fig. 2 . The spatial evolution process of Matao Village - must be improved
As do other pictures.
(Line 179) there are two words “whole” in one sentence
The sentences of this manuscript tend to be very long which makes it hard to be understood by journal readers.
Extensive and considerable editing especially to make sentences as well as paragraphs shorter and more understandable for journal readers.
Data sources are not in the reference.
How does the content of “5.3. Policy implication” relate to the analysis, data source, and research framework? A diagram to explain these relations needs to be created as part of the research method.
The conclusion would be better to include limitations and weaknesses of the study process, especially the methods chosen.
Reviewer 2 Report
In this article the authors carry out a good description of the situation of the locality in question and an adequate application of the chosen spatial syntax model to analyze the circumstances related to the tourist situation that it presents. In addition, all this is done in a clear and perfectly understandable way for the reader and complying with the formal requirements of a research article.
Author Response
-Reviewer 2-
In this article the authors carry out a good description of the situation of the locality in question and an adequate application of the chosen spatial syntax model to analyze the circumstances related to the tourist situation that it presents. In addition, all this is done in a clear and perfectly understandable way for the reader and complying with the formal requirements of a research article.
Response: Thank you for your evaluation and affirmation, which is very important to us. This encouragement strengthens our confidence to continue our research. We will continue to study the spatial evolution pattern of the village. We hope to get your review and suggestions in the future.

Reviewer 3 Report
Dear Authors,
The manuscript presents the results of interesting studies. The results are presented in an interesting way and richly illustrated.
In my opinion, slight adjustments can be made.
Remarks:
1. The abstract should be a total of about 200 words maximum. It is much longer in this manuscript.
2. No research hypotheses / research questions were formulated. In addition, the methodological part (Section 2. Research area and research methods) lacks justification for the use of ‘case study’ for this type of research.
3. The aim of the research should be placed in the Introduction and also in the abstract.
4. There is no DOI in the references
Author Response
Dear Editors,
On behalf of my co-authors, we thank you very much for giving us an opportunity to revise our manuscript, we appreciate editor and reviewers very much for their positive and constructive comments and suggestions on our manuscript entitled “Spatial Evolution Analysis and Spatial Optimization Strategy of Rural Tourism Based on Spatial Syntax Model-- a case study of Matao village in Shandong Province, China”. We sincerely appreciate the thorough review provided by the reviewers. A full account of our responses to the comments from the reviewers· is presented below, and necessary revisions have been made to the manuscript accordingly. Additions and modifications have been marked in red in the revised manuscript. Thank you very much.
***********************************************************
-Reviewer 3-
The manuscript presents the results of interesting studies. The results are presented in an interesting way and richly illustrated. In my opinion, slight adjustments can be made.
- The abstract should be a total of about 200 words maximum. It is much longer in this manuscript.
Response: We appreciate your suggestions very much. We rewrite the abstract section and limit its word number to 198. (Page 1, lines 4-17 ).:
As the tourism industry is embedded in the countryside, space, which is an important means of production of modern industries, has undergone significant changes in the way and trend of spatial transform. This paper aims to analyze the development pattern of rural economic and social structure driven by tourism industry. Based on the spatial syntax model, it takes Matao village of China as the research sample. It decomposes the evolution process of rural space transform: ① in the rural self-driven development stage, the rural space presents a polar core development trend with high understandability and obvious agglomeration. ② in the tourism-driven development stage: Firstly, the driving force of tourism industry to rural space is relatively weak, and the old core still occupies an absolute advantage in the whole village space. With the further development, the spatial scope of high integration and choice values has expanded, resulting in multiple spatial development agglomeration centers. Based on this, it is proposed that the village with tourism industry should integrate the village spatial development planning and landscape style, focus on the regeneration and development of the old space core, and strengthen the integration and correlation of the old and new space clusters.
- No research hypotheses / research questions were formulated. In addition, the methodological part (Section 2. Research area and research methods) lacks justification for the use of ‘case study’ for this type of research.
Response: We appreciate your suggestions very much. We add the necessity of the research on the spatial pattern evolution of this village. (Page 7, lines 194-203).
Oriented to the development process of Matao village, we find that the development of rural tourism industry basically focuses on space construction and space expansion. The breadth and ability of space construction determine the level of rural tourism industry development. According to the stage characteristics of space production, we can more clearly sort out the stage characteristics of the development of rural tourism industry. Based on the research and analysis of the characteristics in different stages of development, it can also explore the problems of space production of tourism industry and put forward strategic suggestions for space optimization. Therefore, the research takes rural space as an important factor of tourism industry production, and introduces the spatial syntax model to analyze the current situation of spatial production in in different development stages.
- The aim of the research should be placed in the Introduction and also in the abstract.
Response: We appreciate your suggestions very much. We rewrite the research objectives of this paper and put it in the introduction and abstract parts. In this modification, we are divided into two parts:
Firstly, in the introduction part.
To address the research gap in existing studies, this paper analyzes the development pattern of rural economic and social structure driven by tourism industry based on the evolution of spatial morphology. The research objectives of this paper are as follows: ① deconstructing the spatial elements and spatial structure, and describes the historical process of rural spatial and temporal evolution in detail quantitatively;② combing the characteristics cognition, model induction and evolution history of spatial transformation driven by tourism industry in detail. ③ analyzing the development law of rural economic and social structure driven by tourism industry and explores the optimization path of rural spatial transformation by the evolution of spatial form. (Page 5, lines134-141 ).
Secondly, in the abstract part.
As the tourism industry is embedded in the countryside, space, which is an important means of production of modern industries, has undergone significant changes in the way and trend of spatial transform. This paper aims to analyze the development pattern of rural economic and social structure driven by tourism industry.----- (Page 1, lines 4-17 )
- There is no DOI in the references
Response: We have added the DOI of each reference and checked the specific information of each reference. Thank you for your advice. In the follow-up study, we will also pay attention to the DOI information of the references. (Page 22-27, lines 580-822 )

Round 2
Reviewer 1 Report
Thank you for the manuscript revision.
I suggest to conduct the proof reading for the second time. For example in line 186 - 196, an obvious proof reading is needed.
Author Response
-Reviewer 1-
- I suggest to conduct the proof reading for the second time. For example in line 186 - 196, an obvious proof reading is needed.
Response: We appreciate your suggestion. We are sorry for the poor language of the reviewed manuscript. We have asked professional proofreader to modify the language. In particular, we have re-edited the part that you suggested to be modified.
‘According to the availability of research data and the characteristics of development stages, this study takes 2003 as the starting point of research, and selecting four years of satellite remote sensing images in 2003, 2008, 2015 and 2020 as the research data map (Data from: https://www.resdc.cn/, accessed on 15 May 2003; 2 June 2008; 27 May 2015 and 24 May 2020). The axis model of space syntax model is drawn by AutoCAD software. In the process of data vectorization, this paper takes the rural village as a whole continuous space, which means that local residents and tourists can follow the natural action law to freely travel within the space. Based on this principle, we remove the closed and confidential area, and the internal road of the scenic spots, and residents` courtyards, so that all the lines on the axis model could be set the same weight to ensure the homogeneity and continuity of the sample space.’ (Page 7, lines195-204).

Reviewer 3 Report
Thank you very much for taking into account almost all my comments. Unfortunately, remark 2 was only partially taken into account. Still no research hypotheses / research questions were formulated.Author Response
-Reviewer 3-
- No research questions were formulated.
Response: We appreciate your suggestions very much. We re-writer the research questions in the introduction part. (Page5, lines 140-154).
To address the research gap in existing studies, This study focuses on three research questions: ① With the embedding of the tourism industry, how does the rural livelihood space evolve and develop, and how does the structure of original living space and labor space transform into tourism industry space ? ② How to realize the analysis and cognition of spatial elements, and to analyze the distribution logic and law of rural spatial elements ③ How to combine the abstract spatial elements with the process of economic and social development, so as to recognize the influence of the construction of spatial elements on the rural economy and society. On account of the evolution of spatial morphology, this paper analyzes the development pattern of rural economic and social structure driven by tourism industry based on the evolution of spatial morphology. The research objectives are as follows: ① deconstructing the spatial elements and structure, and describebing the historical process of rural spatial and temporal evolution in detail quantitatively;② combing the characteristics cognition, model induction and evolution history of spatial transformation driven by tourism industry in detail; ③ analyzing the development law of rural economic and social structure driven by tourism industry and exploring the optimization path of rural spatial transformation by the evolution of spatial form.
